# Phytoestrogens: A Review of Their Impacts on Reproductive Physiology and Other Effects upon Grazing Livestock

**DOI:** 10.3390/ani12192709

**Published:** 2022-10-09

**Authors:** Jessica Wyse, Sajid Latif, Saliya Gurusinghe, Jeffrey McCormick, Leslie A. Weston, Cyril P. Stephen

**Affiliations:** 1School of Agricultural, Environmental and Veterinary Sciences, Charles Sturt University, Wagga Wagga, NSW 2650, Australia; 2Gulbali Institute for Agriculture, Water and the Environment, Charles Sturt University, Wagga Wagga, NSW 2650, Australia; 3National Life Sciences Research Hub, Faculty of Science and Health, Charles Sturt University, Wagga Wagga, NSW 2650, Australia

**Keywords:** phytoestrogen, coumestans, isoflavones, cattle, legumes, reproduction

## Abstract

**Simple Summary:**

Phytoestrogens are secondary plant metabolites that play a role in plant defense, and when ingested by livestock have numerous functions related to reproduction, metabolism, immunological functions and livestock growth and performance. Phytoestrogens are found across various plant species, with the most biologically active of these, isoflavones and coumestans, abundant in legume species. Understanding the overall potential health and reproductive effects that may occur in livestock grazing phytoestrogenic pastures is essential to mitigate any potential risks attributed to fertility loss or to introduce proactive management strategies to aid in improving growth and development. A review of their interactions with livestock systems will provide updated information for the agricultural and veterinary industries.

**Abstract:**

Legume crops and pastures have a high economic value in Australia. However, legume species commonly used for grazing enterprises have been identified to produce high concentrations of phytoestrogens. These compounds are heterocyclic phenolic, and are similar in structure to the mammalian estrogen, 17β-estradiol. The biological activity of the various phytoestrogen types; isoflavones, lignans and coumestans, are species-specific, although at concentrations of 25 mg/kg of dry matter each of the phytoestrogen types affect reproductive functions in grazing livestock. The impacts upon fertility in grazing livestock such as cattle and sheep, vary greatly over length of exposure time, age and health of animal and the stress stimuli the plant is exposed to. More recently, research into the other effects that phytoestrogens may have upon metabolism, immune capacity and growth and performance of grazing livestock has been conducted. Potential new benefits for using these phytoestrogens, such as daidzein and genistein, have been identified by observing the stimulation of production in lymphocytes and other antibody cells. Numerous isoflavones have also been recognized to promote protein synthesis, increase the lean meat ratio, and increase weight gain in cattle and sheep. In Australia, the high economic value of legumes as pasture crops in sheep and cattle production enterprises requires proactive management strategies to mitigate risk associated with potential loss of fertility associated with inclusion of pasture legumes as forages for grazing livestock.

## 1. Introduction

As the focus shifts in Australia and also globally to low-input sustainable agriculture practices, the incorporation of pasture legumes in crop and pasture rotations continues to reduce input costs associated with the application of synthetic nitrogen fertilizer [1]. The addition of forage legumes to perennial pastures can increase the overall nutritional quality of a mixed pasture by improving the available digestible nutrients and protein for grazing livestock [2]. Forage legumes may also reduce the occurrence of toxicosis caused by ingestion of grass pastures such as tall fescue or Phalaris via increasing the diversity of pasture species, thereby providing opportunities for competitive grazing selection by livestock [2]. In recent years, major changes in production systems have arisen due to inclusion of pasture legumes in mixed farming systems. However, dominant stands of pasture legumes have created a new set of challenges for agricultural researchers in the form of agronomic, economic, and animal reproductive problems in Australia [3]. In 2017, the lucerne seed industry in Australia had an estimated value of AUD 88 million and represented 80% of the total legume pasture seed market [4]. In contrast, cow and calf production enterprises were valued at an estimated AUD 15.1 billion (2019–2020) [5]. In production enterprises, high fertility is valued as one of the key economic traits to an efficient herd or flock [6]. In Australia, the high economic value of both lucerne as a pasture crop and in cattle production enterprises requires proactive management strategies to mitigate risk linked to the potential loss of fertility associated with the inclusion of pasture legumes as forages for grazing livestock.

Research on secondary metabolites in crops and pastures has focused on plant defense and the production of phytoestrogens [7]. While phytoestrogens are generally not considered to be essential for plant survival, they are associated with plant protection through plant-to-plant or plant–microbial interactions [8]. More specifically, phytoestrogens are defined as plant-produced, non-steroidal compounds that exert estrogenic effects upon an animal’s central nervous system, inducing estrus and stimulating growth within the genital tracts of females [5]. Phytoestrogens can be classified into distinct classes, isoflavones (being the majority), coumestans and lignans.

As heterocyclic phenolic compounds, phytoestrogens have a similar structure to mammalian estrogen and may exhibit estrogenic or antiestrogenic effects [8]. Phytoestrogenic compounds have now been isolated in over 300 plant species [9], however, few are ingested by humans and/or livestock [10]. Biological activity of various phytoestrogens is highly species-specific [11]. Estrogenic activity has typically been associated with ingestion of forage legumes. Phytoestrogens are also considered as ‘natural’ selective estrogen receptor modulators (SERMs) in mammals and can be characterized by the tissue specific action on estrogens receptors (ERs), permitting the ability to selectively inhibit or stimulate estrogen-like action [12].

Both negative and positive effects from overexposure to phytoestrogen-containing diets in livestock have been observed. The intake of phytoestrogens has been shown to result in embryonic loss, poor semen quality, progesterone deficiency and inhibition of estrus in numerous domestic and non-domesticated species [13]. However, beneficial gains from estrogenic diets, such as increased growth rates and weight gains, have also been recorded in livestock [13,14].

In a recent investigation assessing the concentrations of six key phytoestrogens in commercially available lucerne cultivars, concentrations of individual phytoestrogens ranged from 66.5 to 333.8 mg/kg DM [15]. Currently, producers are recommended to remove or dilute the level of estrogenic feed or forages in the diet of grazing livestock to avoid adverse effects on reproduction [16]. This review seeks to understand and assess the current known effects of phytoestrogens in livestock on metabolism, immune capacity, growth, and reproductive physiology.

## 2. Phytoestrogens

### 2.1. Isoflavones

Isoflavones are currently the most studied group of phytoestrogens [17]. They are present as either aglycons or as glycosides [18], acting as phytoalexins in plants and endogenous hormones in mammals. As secondary metabolites, isoflavones mediate essential interactions between the plant and microbes, including nitrogen sequestration through legume/rhizobium symbiosis, nodulation and plant defense responses. Multiple isoflavones can be found in the same plant species at the same time [18]. Concentration levels of isoflavones have been observed to increase in response to stimuli such as climate (temperature, rainfall, humidity), management (harvest and post-harvest processing) and plant health (soil fertility, pathogen/pest/disease presence) [18]. The in planta concentration of isoflavones is dependent on plant part sampled, stage of growth, cultivar, growing season conditions and sample preservation after collection or harvest [18]. Isoflavones may promote vasodilation in tissues, reducing the effect of vasoconstriction caused by ergot alkaloids in ruminants [2]. Isoflavones also serve as precursors for other isoflavone phytoalexins for fungistatic, antibacterial, antiviral and antioxidant properties.

Clover disease, first identified in Australia in the 1940s, was determined to be induced by high concentrations of isoflavones, in particular, formononetin in clover varieties [18]. After the discovery of phytoestrogens in clover, and the resulting reproductive failures in sheep, breeding objectives developed by forage breeders focused on reducing phytoestrogen concentrations in new clover varieties, with a focus on isoflavones. Isoflavones typically occur in plants in the form of glycosides, which can be further metabolized to increasingly complex classes of molecules, including pterocarpans and coumestans [19]. Isoflavones and their resulting metabolites account for ~1–5% of the phytoestrogens noted in the blood plasma of cattle and sheep in their unconjugated forms [19]. Daidzein (Figure 1a), formononetin (Figure 1b), genistein (Figure 1c) and biochanin A (Figure 1d) have been identified to be the key isoflavones found in legume and other fodder consumed by cattle, sheep and horses [20].

Both genistein and daidzein, like 17β-estradiol, are considered biphasic [21]. At low concentrations both can be stimulatory of animal growth. However, at higher concentrations they can be inhibitors of animal reproductive functions. Genistein is a potent inhibitor of tyrosine kinase in addition to its estrogenomimetic properties [21]. Equol (Figure 1e), although included in the isoflavone class as a metabolite of daidzein, is not a naturally produced plant compound. Rather, it is a metabolic product of intestinal bacteria [18] first isolated from the urine of pregnant mares in the early 1930s [18].

### 2.2. Coumestans

Up until the early 1950s, lucerne (*Medicago sativa*) and white clover (*Trifolium repens* L.) were classed as non-estrogenic legume forages, however, it was later established that both species exhibit frequent and significant estrogenic potency [22]. Research by the Western Utilization Research and Development Division (WURDD) in 1955 isolated a new benzofurocoumarin derivative from white clover, which was later recognized as a new class of phytoestrogens; coumestans [23]. These compounds were later identified to be biogenetically related to isoflavones. Biotic stimuli such as viral, bacterial and fungal pathogens are known to increase the formation of numerous aromatic compounds, including coumestrol and its metabolites as key defense molecules [24]. Although the stimulation of the production of coumestans can be dependent on foliar pathogens, other factors such as the stage of growth, selection of cultivar [25] and soil type [26] also contribute to coumestan production.

In comparison to isoflavones which have a limited impact on estrous cycles, coumestans have been identified to have greater potential for the inhibition of estrous [27]. Several coumestans are currently characterized as having estrogenic potency [28] and these are commonly synthesized in large quantities in legumes that have been infected by various fungal pathogens [29]. The most important of these coumestans is coumestrol (Figure 2a), which has largely been established to be synthesized in legumes [28]. Isolated originally from *Trifolium repens* L. [23], it has also been found in low concentrations in other clover species such as in *Trifolium pratense* [30].

Coumestrol has been identified as the dominant phytoestrogen in lucerne, white clover and some annual medics [22]. Concentrations of coumestrol in lucerne have been recorded to range from 0 to 100 mg/kg DM, although coumestrol content can be upregulated by over 180-fold, in response to pathogen-induced stimuli [24]. Coumestrol concentrations in vitro are typically high in estrogenic potency, similar to that of 17β-estradiol [31]. Coumestrol acts similarly to 17β-estradiol when administered exogenously, subsequently resulting in ovulation failure [31]. Considered the next most important coumestan after coumestrol, 4′methoxycoumestrol (Figure 2b) is often detected in higher concentrations than coumestrol in some species of annual medics such as *Medicago littoralis* [27,32]. Other notable coumestans include 3′methoxycoumestrol, trifoliol, sativol, repensol, lucernol, medicagol and 11,12-dimethoxy-7-hydroxycoumestan. Although, these are generally found in low or undiscernible concentrations in clover and lucerne extracts, unless production is stimulated by foliar or fungal pathogens, and subsequent concentrations are then easily detectable [33]. Recently, 3′methoxycoumestrol has been quantified in bovine plasma after a grazing period of 21 days on lucerne (var. Genesis) where concentrations increased significantly over the period [15].

### 2.3. Lignans

Lignans, classed as non-flavonoids, are found in numerous vascular plant species [34]. There have been several hundred lignans identified, although exact numbers are not likely accurate due to highly complicated analysis required to elucidate the structures of glycoside forms [34]. Lignans are bioactive compounds that act as antioxidants and anti-inflammatories, among other biological properties, that are important in plant defense against pests and pathogens [35]. They are also essential for plant development and structure. Lignans are predominantly present in plants as free forms, with their glycosides present in the minority of those elucidated molecules [35]. Lignans possess great structural diversity, and consist of two phenylpropane units as the molecular backbone of their structure [35]. Secoisolariciresinol (Figure 3c) and matairesinol (Figure 3d) have been commonly isolated in cereals, oilseeds, and legumes. Enterodiol (Figure 3a) and enterolactone (Figure 3b) are metabolites of secoisolariciresinol and matairesinol, known as ‘enterolignans’ or ‘mammalian lignans’ [34]. These metabolites are formed by the action of gut microflora upon lignans.

Enterolignans though research have been observed to display greater antioxidant activity than vitamin E [36]. There is a proposed application for these bioactive compounds for use in alleviating oxidative stress and immune depression in newborn calves [36]. As well as having antioxidant and anti-inflammatory properties, the enterolignans also act as both estrogen agonists and antagonists [37].

## 3. Metabolism and Metabolic Effects

Phytoestrogens, which are predominantly present as glycosides in plant material, are not estrogenically active [38]. Demethylation, methylation, hydroxylation, chlorination, iodination and nitration of phytoestrogens generally occurs after ingestion by livestock [13]. Phytoestrogens and their resultant metabolites are predominantly found in plasma in both conjugated and sulfo-conjugated forms [39]. They are converted in vivo to more estrogenically active unconjugated forms via metabolism (Figure 4) [39].

After ingestion of the glycoside forms within the plant material, the metabolites are largely absorbed [39] and hydrolyzed in the rumen. Biochanin A and genistein is transformed to p-ethyl phenol, whilst formononetin and daidzein transform into equol [13]. Conversion of the lignan secoisolariciresinol to enterodiol and enterolactone also occurs in the rumen [36]. Njåstad (2011) observed a high metabolism of phytoestrogens in the rumen and after 6–10 days, the rumen microbes were more adapted to metabolizing phytoestrogens [38]. Conjugation of these compounds to carbohydrates, producing an aglycone phytoestrogen, results in a bioactive form [13]. Hydrolysis also occurs in the small and large intestines (more so in non-ruminant livestock species) [31] via β-glycosidase enzymes produced by either gut micro-organisms or intestinal epithelial glands [13], before being absorbed into the blood.

Sheep have greater conjugative activity than cattle in most parts of the gastrointestinal tract, except for in the small intestine [36]. Re-conjugation of free phytoestrogens with glucuronic acids and sulfuric acid [13,19], along with further metabolism, occurs in the liver and kidneys [28]. Enterohepatic recirculation of these metabolites occur post-ruminally [40]. Phytoestrogens can then be transferred into the milk of lactating ruminants [41], and this transferal is directly correlated with the intake of estrogenic feed [38]. The excretion of these phytoestrogens occurs through urine, in conjugated forms and in feces unconjugated forms [28,41]. Indirectly, the ingestion of phytoestrogens reduces the amount of biologically active estradiol present in serum, via the stimulation of the hepatic synthesis of steroid hormone binding globulin (SHBG) [42]. This results in isoflavones displaying weak antiestrogenic effects in the absence of estrogen, and strong antiestrogenic effects in the presence of estrogen [42].

Few studies have been performed on the metabolism of coumestans in both sheep and cattle, focusing on phytoestrogen metabolism concerning the ingestion of isoflavones and lignans. It is now well accepted that the metabolism of coumestans occurs within the rumen for both species. Braden and McDonald (1970) identified the demethylation of 4′methoxycoumestrol to coumestrol occurring in the rumen of ewes, via rumen microbes, with the rate of demethylation dependent on enzymatic activity [43]. 4′methoxycoumestrol has also been observed to have greater estrogenicity after demethylation in the rumen, allowing for the hydroxyl groups to bind to the estrogen receptors [44]. Unlike other phytoestrogens, coumestrol is considered estrogenic without requiring to be metabolized [38].

Phytoestrogens typically bind to estrogen receptors (ERs) in mammals where they are then involved in several regulatory processes. Their binding affinity to ERs is in the following order: 17β-estradiol > coumestrol > genistein > equol > daidzein > biochanin A [17]. Coumestrol has been observed to act similarly to estradiol (E_2_) when administered exogenously. Subsequently, livestock have been observed failing to ovulate due to disruption to follicular growth [31]. Coumestrol binds to the ERs which induces the expression and regulation of numerous processes that relate to growth. The two main subtypes of ERs that phytoestrogens bind to are ERα and ERβ [45,46]. Coumestrol acts as an antagonist to ERα and ERβ, however, in comparison, its binding affinity for ERα is weaker than for ERβ [47]. Coumestrol also affects the expression of androgen, glucocorticoid, progesterone and thyroid receptors [47]. Transcription is not the only means of disrupting hormone function, as coumestans also modulate genomic effects of endogenous E_2_ by binding to or inactivating enzymes which are important for E_2_ biosynthesis.

## 4. Effects on Antioxidant and Immune Capacity

Phytoestrogens, largely the isoflavone class, have been observed to influence antioxidant and immune capacity in cattle [48]. Daidzein (an isoflavone) has been observed to significantly increase serum levels of immunoglobulin G (IgG), M (IgM) and A (IgA) [49]. Bull calves who received 200 and 400 mg/kg of daidzein were found to have greater immune activity when compared to a control group, suggesting that daidzein has the potential to improve immune function [49]. This was not observed to have a significant effect on two-year-old steers at concentrations of 500 and 1000 mg/kg of daidzein [50]. However, age, breed of cattle and methods of feed administration should be taken into consideration when comparing antioxidant and immune capacity. Estrogen plays a role in the function of the growth hormone (GH)/insulin growth factor 1 (IGF-1) axis. Exogenous estrogens, such as daidzein, can bind to and activate estrogen receptors, resulting in augmented effects, such as significantly increased serum concentrations of both GH and IGF-1 secretion [49]. Daidzein has also been found to have a positive effect upon the hypothalamic–pituitary–adrenal (HPA) axis which aids in the regulation of reproductive functions and nutrition [51]. It has also been suggested that daidzein could potentially stimulate the activation and production of lymphocytes and production of other antibody cells [49].

Genistein, another isoflavone, has also been shown to significantly inhibit the replication of the bovine herpes virus 1 (BHV-1) [52], which results in infectious bovine rhinotracheitis (IBR), a highly contagious respiratory disease that affects both young and old cattle. The inhibition of BHV-1 and other alphaherpesviruses via genistein affects the phosphorylation of protein kinases, such as tyrosine [52]. A dose of genistein (25 µM) was observed to reduce viral replication by almost 90%, 18 h after inoculation [52], although the reduction was not sustained at 24 h. Studies investigating the potential use of genistein as an anti-viral agent at greater concentrations through orally administered legume feeds in vivo are underway [52]. Genistein has also been observed to inhibit the bovine viral diarrhea virus (BVDV) infection, a pestivirus which can result in fatal mucosal disease [53], in a dose-dependent manner. The inhibition of BVDV by up to 50% at concentrations between 50 and 100 μg/mL genistein has been observed. Although genistein was a weaker inhibitor during the early stages of infection, inhibition during the later stages of the infection was higher [53].

## 5. Effects on Growth and Performance

There are several phytoestrogens that have been identified to have beneficial effects, such as stimulating growth rate and increasing weight gain in livestock. In particular, a high genistein: low formononetin ratio has been observed to increase the liveweight gain in ewes and lambs when fed subterranean clover [54]. This was also correlated with a higher uptake of metabolizable protein. The average daily gain (ADG) in the ewes and lambs after the 8 week feeding period was higher by 26.3% and 31.7%, respectively, compared to the control animals consuming Italian ryegrass [54]. Carcass weights were also found to be 18.7% higher than that of those in the control group. Other observations included greater internal and total fat percentages, longer and heavier leg bone and earlier maturation of treatment animals [54]. This research suggests that certain phytoestrogens such as genistein may be effective anabolic agents, however, this effect may have been masked by the addition of supplementary feeding.

This would be useful as an alternative method for promoting growth in feedlot operations. Feedlot operators may utilize growth hormone injections in cattle to improve eating quality, by decreasing the accumulation of fat and increasing the ratio of lean meat in young heifers and steers [55]. This in turn increases feed efficiency, allowing greater growth to be achieved on equal to or reduced quantity of feed. This also results in the reduction of costs for both producers and consumers. Isoflavones have been identified to promote growth in grazing and feedlot animals, minimizing the time required to achieve target weights. Of these, zearalenol (a naturally produced fungal metabolite bi-product of zearalenone) is widely used as a growth promoter in the United States [56]. Zearalenol and α-Zearalanol have been found to promote protein synthesis, increase the lean meat ratio, and increase weight gain in cattle and sheep similarly to other estrogenic growth hormones in diets [56]. α-Zearalanol, a metabolite of zearalanol, however, has been found to be twice as effective as zearalanol, and less toxic [56].

The isoflavone daidzein has also been observed to be beneficial in promoting the digestion of dietary proteins and improving the secretion of growth hormones, thus, resulting in the improved physiological condition in bull calves and thereby enhancing production [49]. Runyan et al. (2018) observed the potential effect of soybean meal (used as an alternative protein source for bull growth) upon bull fertility at elevated levels in the diet for an extended period. Bull calves achieved a higher average daily gain (ADG) when fed 10% soybean meal diet than those of the same age group on cotton seed meal. The supplementation of daidzein into the diet has also demonstrated the enhancement of fermentation within the rumen of beef cattle [57]. Unlike zearalanol, daidzein has been observed to increase the intramuscular fat content and marbling score in steers [50]. This is as a result of enhancement of lipid synthesis due to the estrogenic activity of daidzein, which is considered to play an inhibitory role in the β oxidation of fatty acids in the liver, leading to elevated serum levels of free fatty acids [57]. However, there has been no consistent evidence to support these findings across all breeds, ages and sex of cattle.

## 6. Effects on Female Reproductive Physiology

Temporary reversible or potentially permanent infertility can result without any evident signs, and may only be detected through measurement of concentrations within the diet or plasma, or observable by clinical effects [58]. Exposure to a highly estrogenic diet during critical development periods, such as pre-puberty or gonadal maturity, may result in detrimental impacts upon future reproductive functions and fertility in livestock [59]. The effects of various phytoestrogens upon the reproductive tract of females in different species is highly dependent on factors such as age and duration of exposure [42]. Additionally, the relative estrogenic potency of an individual phytoestrogen is dependent on the type of assay used to measure activity, route of administration (ingestion versus injection), animal species, duration of exposure and timing of exposure [60]. The effect of exposure to phytoestrogens on individual animals varies based on duration of exposure and stage of reproductive development [16].

### 6.1. Cows

Phytoestrogens can be characterized as biphasic, as both positive and negative effects can be associated with reproductive functions, dependent on the dosage and reproductive status of the cow [61]. In both cows and heifers, many negative reproductive impacts have been recorded, although they are broadly regarded to be a result of phytoestrogens as a collective, rather than focused on coumestans (Figure 5). These effects include hypertrophy of the duct epithelium, temporary infertility (which is usually resolved after one month from the removal of estrogenic feed), increased secretion of fluid, enlargement of the uterus, nymphomania [44] and other clinical signs resembling cystic ovaries [58].

Phytoestrogens have been demonstrated to act both acutely and chronically upon the reproductive tract of cattle [62]. It has been reported that cattle are less sensitive to the effects of isoflavones than sheep [63]. In cattle, isoflavones have been observed to inhibit estrus signs, via the inhibition of the in vitro activity of aromatase, an enzyme essential for estrogen biosynthesis and ovarian follicular development [63]. Piotrowska [62] demonstrated that higher concentrations of the isoflavone metabolites equol and *p*-ethyl phenol (>4 and >40 nmol/g) in the plasma and tissue of the corpus luteum (CL), at insemination and early stages of pregnancy, can be associated with embryonic loss at the early stages of pregnancy in heifers on a high soy diet compared to the control. A decreased rate of conception in heifers along with an increased mean rate of insemination was also observed.

Coumestans have also been noted to stimulate earlier development of mammary glands and genitals in heifers [64,65]. Vaginal prolapse has also been observed in heifers at coumestrol concentrations greater than 37 mg/kg DM, with some individuals experiencing prolapse numerous times [64]. Resumption to estrus after temporary infertility caused by coumestans can be attained through the treatments for cystic ovaries [44]. Phytoestrogens have also been observed to decrease the rate of oocyte maturation in cattle [66]. Coumestrol and genistein have both been observed to increase the number of immature oocytes at concentrations of 10 and 100 µg/mL, respectively [66]. Although coumestrol was found to inhibit the maturation of oocytes at concentrations of 10 µg/mL, genistein inhibited oocyte maturation when concentrations were greater than 100 µg/mL [66].

### 6.2. Ewes

Research conducted by Adams [67] observed the changes in the reproductive tract in non-pregnant ewes on an estrogenic cultivar of sub-clover (cv. Dinninup). Isoflavones (genistein, formononetin and biochanin A) present in ewes grazing sub-clover were associated with greater numbers of follicles in the ovaries, with many being deficient in antrum formation leading to early follicular atresia (Figure 6). Subacute endometritis and edematous uterine submucosa were observed in the uterus and squamous metaplasia and increase in cervical crypt complexity were also observed in the cervix of the ewes [67]. Other clinical signs such as endometrial cysts, as well as ecchymoses and petechiae in the mucosa of the uterus, have been observed by Cantero et al. [68] in ewes grazing lucerne, where the concentration of coumestrol ranged from 17 to 30 ppm. The cervix uteri also had cystic glandular hyperplasia and hypertrophic endometrial folds [68]. Two of the ewes were also observed to have paratubal cysts. Ewes grazing lucerne were recorded to have greater glandular activity in the endometrium than those in the control group. Prolapse and pyometra have also been recorded in ewes resulting from prolonged exposure to estrogenic feeds [42].

Ewes also experience similar negative reproductive effects to those of cattle, however, defeminization of the cervix has been observed in ewes after prolonged exposure to estrogenic feed which results in permanent infertility [58]. Defeminization of the cervix occurs after functional redifferentiation and loss of sexual features [42]. In sheep, the genes that express sexual differentiation are not entirely deactivated at birth, as a result defeminization occurs [42]. This also results in the reduced ability of the cervix to store spermatozoa, resulting in reduced conception/lambing rates, although the ovarian function in the ewe remains normal [58].

Concentrations of coumestans as low as 25 ppm can significantly decrease the ovulation rate of ewes, and ewes previously recorded to have high ovulation rates have been determined to be significantly more sensitive to coumestans than ewes with lower ovulation rates, when grazing lucerne [69]. High concentrations of formononetin in ovariectomized ewes have been observed to change teat length and color of the vulva, while uterotrophic effects have been observed in ewes grazing red clover silage [70]. In more severe cases, prolapse of the cervix, vagina and the rectum have occurred in ewes, in response to high concentrations of formononetin in subterranean clover [54]. Return to estrus in the ewe after exposure to lucerne pasture with moderate estrogenicity (29 mg coumestrol/kg DM) is almost immediate once removed from the pasture, with two weeks allowing full recovery from any negative effects [71].

### 6.3. Mares

Mare reproductive loss syndrome (MRLS) was identified in 2001 in Kentucky, with phytoestrogens suspected to be a contributing factor to early and late fetal losses [72,73]. Even though analysis identified a broad range of phytoestrogens in the pasture samples consumed, the results of the samplings were inconclusive. Subsequent research comparing mares to cows and ewes identified that mares may also experience temporary infertility, in some instances as rapidly as 14 days following lucerne ingestion [74]. After exposure to lucerne for a minimum of 14 days, lack of ovulation, uterine edema and fluid accumulation were recorded in all eight mares in the trial (Figure 7) [74]. Ferreira-Dias et al. [31] also observed uterine edema and persistent anovulatory follicles in mares fed on a lucerne diet. Two to three weeks after removal from lucerne hay, uterine edema regressed and cyclicity resumed in the mares [31,74]. Coumestrol was also found to increase in vivo and in vitro production and secretion of uterine prostaglandin [74] resulted in reduced levels of progesterone concentration in the blood plasma [74]. Coumestrol was noted to be a stronger stimulator of prostaglandin when compared to 17β-estradiol [74]. The study by Szóstek et al. [74] suggests that phytoestrogens may directly disrupt reproductive efficiency and uterine function by modulating the ratio of prostaglandin F2 alpha:dinoprostone (PGF_2a_:PGE_2_), leading to high, nonphysical production of luteolytic PGF_2a_ during the estrous cycle and early pregnancy. The coumestans, coumestrol and 4′methoxycoumestrol, have been observed in both free and conjugated forms in mare’s plasma after ingesting lucerne feeds [75].

## 7. Effects on Male Reproductive Physiology

Exposure to phytoestrogens during the neonatal period may result in reproductive abnormalities in males such as the downregulation of testicular gene expression [76]. Estrogen is known to hold a role in inducing the acrosome reaction in ram spermatozoa, with the majority of estrogen receptors localized in the sperm head, with higher densities found in the post-acrosomal region. Given that estrogenic compounds, including equol, are shown to reach mammalian seminal plasma, and are able to bypass the placenta and blood–brain barrier, it is possible that spermatozoa may encounter equol upon exposure to seminal plasma or in the female reproductive tract [77]. Specifically, isoflavones are able to directly alter the function of spermatozoa [77]. Low concentrations of isoflavones in vitro have been shown to promote premature capacitation, acrosome loss and inhibition of the acrosomal reaction [77].

### 7.1. Bulls and Steers

There have been negligible recorded reproductive impacts on bull fertility in regard to phytoestrogens in vivo. Phytoestrogens have been reported to contribute to an increase in weight, growth and circumference of the scrotum and higher sperm counts in prepubertal bulls [78] (Figure 8). Runyan et al. (2018) observed the trend of feeding soybean meal (used for an alternative protein source for bull growth) to improve bull fertility and average daily gain, at elevated levels in the diet for an extended period. Semen quality and average daily gain in bulls was increased, although further investigation is warranted into the quantification of the phytoestrogens in the feed, and whether the effect was influenced by the high protein source [79]. Yurrita (2016) also studied the inclusion of soybean meal (containing 10% dietary phytoestrogen) into the diet of bull calves, observing improvement in scrotal growth and semen quality at maturity. This was noted in bulls produced by maiden two-year-old heifers [80]. Minimal incidence of negative reproductive functions in steers was reported (Figure 9), with the exception of a case of hypertrophy of the duct epithelium [58]. Although other effects such as steers exhibiting sexual behavior such as bulling and development of mammary glands (gynecomastia) [64], has been provided from anecdotal evidence [44,81]. These effects can be seen in steers at concentrations of coumestrol of 37 mg/kg DM or greater [64].

Research has also been conducted to investigate the potential of beneficial effects of genistein upon the thawing of cryopreserved semen. Genistein has estrogenic potential to inhibit protein tyrosine kinases (PTK) which plays a key role in numerous sperm functions, including sperm capacitation [82]. Concentrations of genistein below 1 µg/mL have a significant effect upon decreasing sperm–zona pellucida binding and the inhibition of tyrosine phosphorylation [83]. Despite the observations in some other species, concentrations of genistein added to the thawing media did not show beneficial improvements in terms of sperm motility, viability, capacitation, acrosomal reactions (progesterone and ZP3-6 peptide induced) or sperm zona binding, for bovine semen at concentrations ranging from 0.074 to 7.4 µmol/L [82]. The addition of genistein to the thawing process reduced the acrosomal reactions and disrupted the signal transduction pathways which are crucial for sperm–ovum interaction [82]. Further research into the effects of the addition of genistein to sperm has shown a reduction in motility and membrane integrity [84]. Similar to the observed beneficial effects reported in bovine semen, the addition of 10 µM of genistein to cryopreserved ram semen resulted in a significant increase in acrosomal integrity and post-thaw motility [85]. However at concentrations of 100 µM of genistein, there was a reduction in post-thaw motility, suggesting that higher concentrations of genistein may have deleterious effects upon the quality and functional integrity of the ram sperm [85]. Lower concentrations of genistein (<5 µM) were observed not to affect quality or integrity.

### 7.2. Rams and Wethers

Wethers have been recorded to have significant negative physiological impacts due to ingestion of high concentrations of phytoestrogens. Along with hypertrophy of the duct epithelium (Figure 10), wethers have also been found to have enlarged bulbourethral glands and obstruction to the kidneys (due to buildup of sediments) from prolonged exposure to grazing estrogenic pastures [39]. Both have resulted in wether mortalities. Enlarged teats and occasional lactation in wethers has also been observed, with the enlargement of teats being used to measure the estrogenic content of pasture in some instances [58].

A study into the effects of phytoestrogens on the functionality of ram spermatozoa revealed that normal sperm function can be compromised as rapidly as 30 min after exposure to 0.1–1 µM of the isoflavone equol, becoming more pronounced after 6 h of exposure [77]. It was also observed to decrease sperm motility, increase membrane fluidity and increase DNA fragmentation after exposure to 0.1–1 µM of equol after 30 min. It also promoted the mitochondrial superoxide production and promotion of lipid peroxidation [77]. Although both 17β-estradiol, equol and numerous other phytoestrogens promote spermatozoa capacitation and acrosomal reaction, it is suggested that equol may initiate spermatozoa capacitation in rams, resulting in reduced motility.

### 7.3. Stallions

The effect of genistein upon equine semen quality has also been observed, with the addition of genistein to equine sperm (up to 800 µM) not significantly influencing viability [86]. Genistein also did not affect sperm motility or acrosomal status immediately and up to one hour after thawing [86]. It can be concluded that the beneficial effect of genistein on sperm motility is species-specific and may also be breed-specific.

## 8. Conclusions

As a secondary mechanism for plant defense, phytoestrogens have recently received increasing attention due to their potential biological activity in mammals related to reproduction, metabolism and immunological function among other properties. The literature reviewed in this study has focused on the two main livestock species that would typically graze a legume pasture, cattle and sheep, and a third, the horse, that also may commonly ingest legume forages as either a pasture or as hay. The information surrounding phytoestrogens and their role in the plant and/or grazing livestock is still not completely understood, with there being significant gaps in the literature. For producers, understanding the overall potential health and reproductive effects in livestock grazing phytoestrogenic pastures is essential to mitigate any future risks attributed to fertility loss or to introduce proactive management strategies to aid in improving livestock growth and development. Information on current and evolving interactions of these compounds within livestock systems not only aides the research community in looking at further health benefits, but also aids agricultural and veterinary industries in understanding and managing such interactions. This review does not seek to deter producers from the use of pastures that may contain phytoestrogenic compounds but aims to provide updated information on the evolving research into the interactions of these plant metabolites with grazing livestock.

## Figures and Tables

**Figure 1 animals-12-02709-f001:**
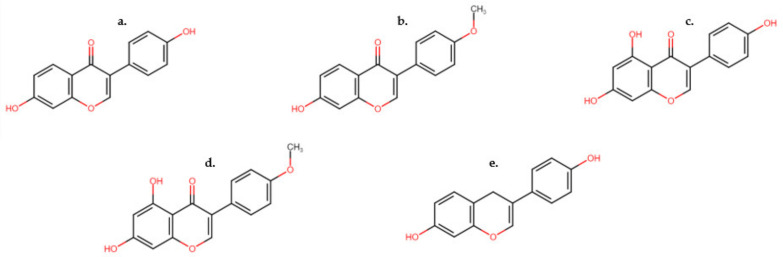
Structures of common isoflavones found in legume species. (**a**) Daidzein, (**b**) formononetin, (**c**) genistein, (**d**) biochanin A and (**e**) equol, a metabolite of daidzein.

**Figure 2 animals-12-02709-f002:**
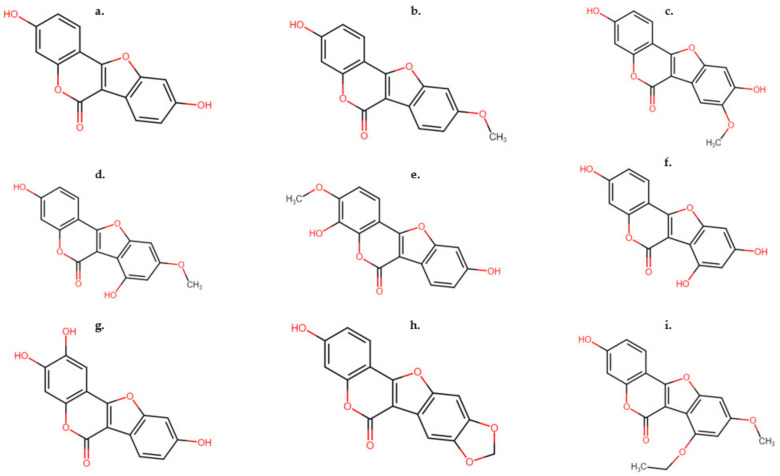
Structures of common coumestans found in legume species. (**a**) Coumestrol, (**b**) 4′methoxycoumestrol, (**c**) 3′methoxycoumestrol, (**d**) trifoliol, (**e**) sativol, (**f**) repensol, (**g**) lucernol, (**h**) medicagol and (**i**) 11,12-dimethoxy-7-hydroxycoumestan.

**Figure 3 animals-12-02709-f003:**
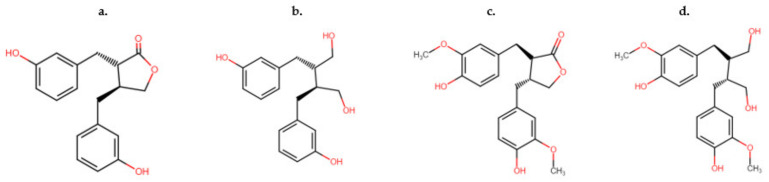
Comparison of the two distinct types of lignans absorbed by ruminants and non-ruminants (**c**) matairesinol, (**d**) secoisolariciresinol, and their metabolites (**a**) enterodiol and (**b**) enterolactone.

**Figure 4 animals-12-02709-f004:**
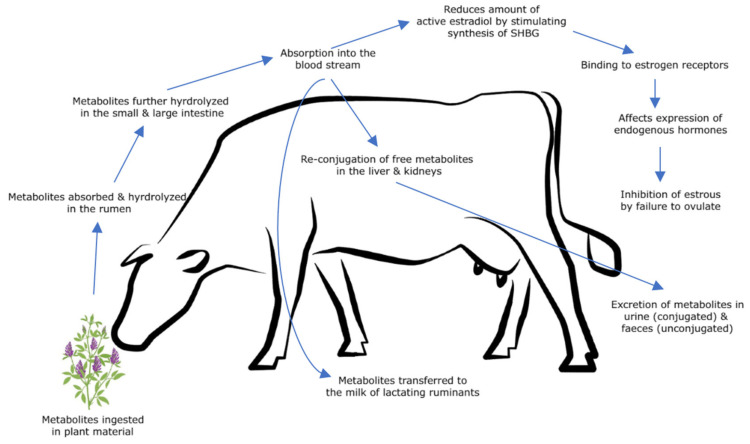
Metabolism of isoflavones and coumestans in grazing ruminants.

**Figure 5 animals-12-02709-f005:**
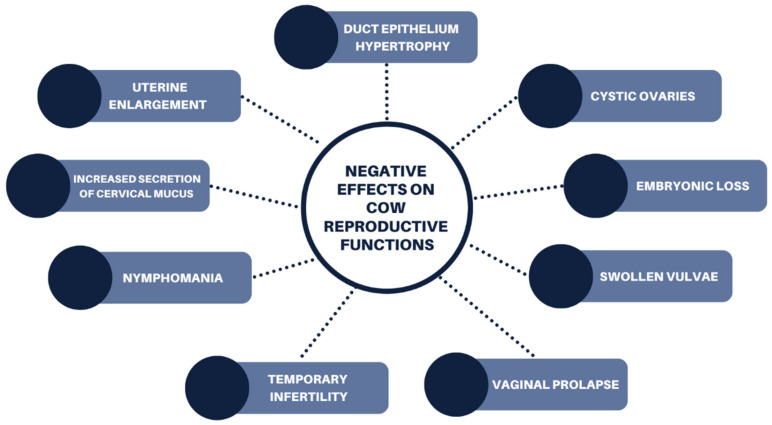
Negative effects of phytoestrogens on cow reproductive functions following ingestion of legume pasture species.

**Figure 6 animals-12-02709-f006:**
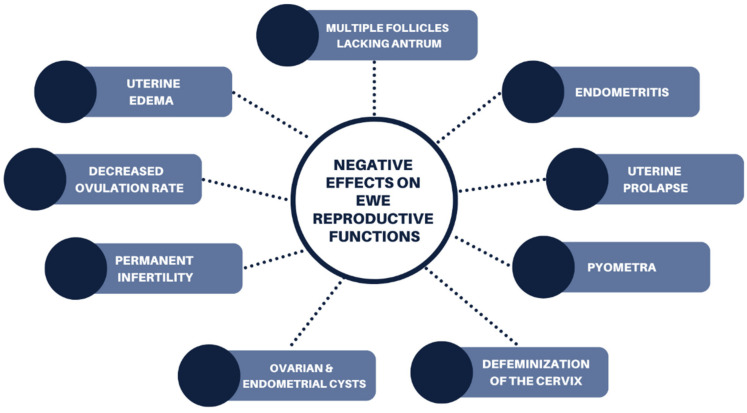
Negative effects of phytoestrogens on ewe reproductive functions from legume pasture species.

**Figure 7 animals-12-02709-f007:**
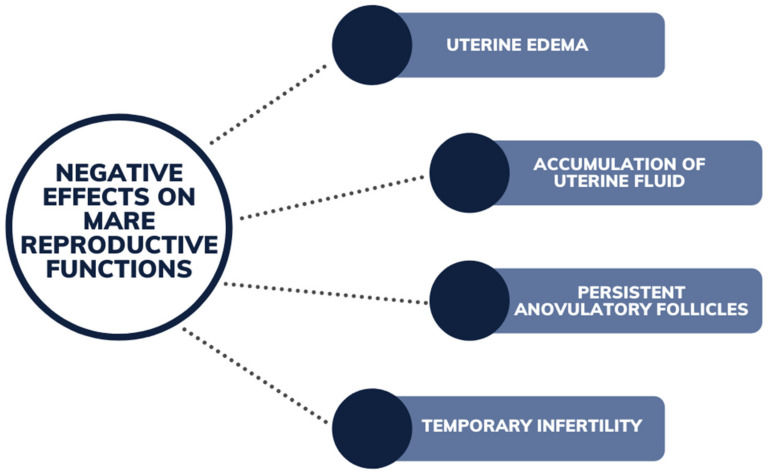
Negative effects of phytoestrogens on mare reproductive functions induced following ingestion of legume pasture species.

**Figure 8 animals-12-02709-f008:**
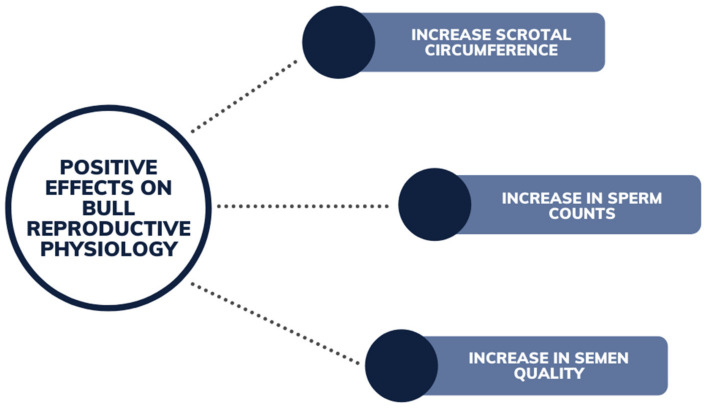
Beneficial effects of phytoestrogens on bull reproductive physiology from grazing legume pastures.

**Figure 9 animals-12-02709-f009:**
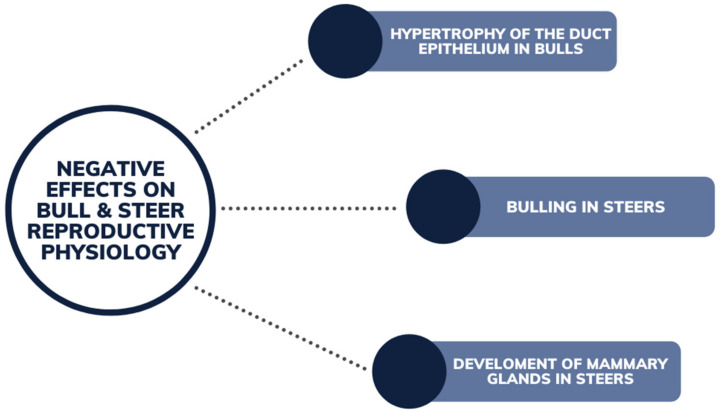
Negative effects of phytoestrogens on bull and steer reproductive physiology from grazing legume pastures.

**Figure 10 animals-12-02709-f010:**
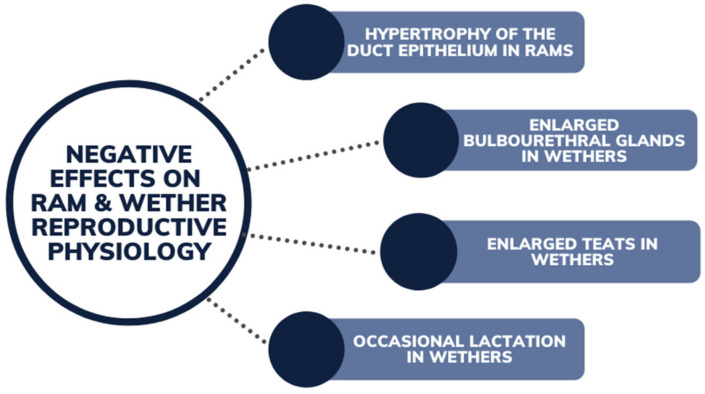
Negative effects of phytoestrogens on ram and wether reproductive physiology from grazing legume pastures.

## Data Availability

Not applicable.

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
