# Peer review of "Phytoestrogens: A Review of Their Impacts on Reproductive Physiology and Other Effects upon Grazing Livestock"

_animals, 2022, doi:10.3390/ani12192709_

Round 1

Reviewer 1 Report

This is a review of an area which is of interest to grazing livestock production. The area has been reviewed extensively in the past and there are some very good fairly recent reviews. This paper, as well as covering the negative reproductive issues with phytoestrogens discusses some potentially positive effects which seem to be recent potentially important developments.

Comments

Introduction (page 2 lines 44 - 56 could be abreviated

Section 2.2 on page 4 line 14 should be dependant on foliar pathogens

Section 3 page 7; I like fig 4

Section 4 page 7 & 8 good.but on the viral inhibition  story, what is the test system and dose?

Section 5 page 8 & 9 good information

section 6.1 figure 5  why is embryonic loss not included?

section6.2 figure 6  why is decreased ovulation rate not included?

section 7.1 &7.2 Figures 8 &9 are from grazing animals but the text indicated both positive and negative in-vitro effects and they seem to be mainly negative

Author Response

  • Introduction (page 2 lines 44 – 56 could be abbreviated - Some of the information has been condensed.
  • Section 2.2 on page 4 line 14 should be dependent on foliar pathogens -This has been corrected, thank you.
  • Section 3, Page 7 – Appreciate your comments, thank you.
  • Section 4 page 7 & 8 good.but on the viral inhibition story, what is the test system and dose? - This information has been included - A dose of genistein (25 µM) was observed to reduce viral replication by almost 90%, 18 hours after inoculation.
  • Section 5, Page 8 -9 – Appreciate your comments, thank you.
  • Section 6.1, Fig 5: please include – embryonic loss - Embryonic loss included.
  • 2 figure 6 why is decreased ovulation rate not included? - Ovulation rate included. 
  • section 7.1 &7.2 Figures 8 &9 are from grazing animals but the text indicated both positive and negative in-vitro effects and they seem to be mainly negative

This has been changed to in vivothank you

The authors would like to thank you for the suggestions/comments. This has helped to improve the quality of the manuscript. Please note the figures are only a quick summary of some of the effects observed within both male and female of that species. They are not a complete listing of all effects.

Reviewer 2 Report

 The manuscript entitled " Phytoestrogens: a review of their impacts on reproductive physiology and other effects upon grazing livestock" is an interesting article. The article needs minor revision before further consideration.

1.     If the "other effects upon livestock" in the caption is specific.

2.     Some parts of the introduction need to be more logical. For example, the article does not describe the content mentioned in lines 82-84.

3.     In the preface, the author explains the advantages and disadvantages of phytoestrogens. However, in the introduction of the three estrogen hormones in the second part, only isoflavones have a small amount of disadvantages, and the other two hardly have any disadvantages.

4.     2. The volume in part 3, 4, 5 and 6 of the article is large, so whether the structure can be considered. The authors reviewed a large amount of literature on the concentration of phytoestrogens at which experimental results are good and bad in Part 2.

5.     Lines 258-273, here isoflavones, isoflavone daidzein, daidzein need to be explained. Isoflavones in the first sentence, isoflavone daidzein in the second, and daidzein in the next. Whether these three are equal, or what kind of relationship they are, requires a brief explanation.

6.     There are few references on phytoestrogens in the last 5 years, and few studies on the effects of phytoestrogens on ruminants.

7.     Part 8 is inconsistent with the overall logic of the article and does not closely follow the title.

Author Response

    1. If the "other effects upon livestock" in the caption is specific.

    It is specific for this article as, in addition to the effects on reproductive physiology, we have listed and discussed the important effects on  metabolism, immune capacity and growth and performance of grazing livestock.

    1. Some parts of the introduction need to be more logical. For example, the article does not describe the content mentioned in lines 82-84.

    Thank you for the comment. I have removed lines 82 – 84 and reworded the last paragraph of the introduction.

    1. In the preface, the author explains the advantages and disadvantages of phytoestrogens. However, in the introduction of the three estrogen hormones in the second part, only isoflavones have a small amount of disadvantages, and the other two hardly have any disadvantages.

    Section 2 is only an introduction to the three main classes of phytoestrogens and their respective metabolites. It is written to provide readers background before reading through the effects phytoestrogens have upon livestock.

    1. The volume in part 3, 4, 5 and 6 of the article is large, so whether the structure can be considered. The authors reviewed a large amount of literature on the concentration of phytoestrogens at which experimental results are good and bad in Part 2.

    Thank you for the comment/suggestion. Not sure if you wanted any changes in this section? I have kept all the information to make it comprehensive.

    1. Lines 258-273, here isoflavones, isoflavone daidzein, daidzein need to be explained. Isoflavones in the first sentence, isoflavone daidzein in the second, and daidzein in the next. Whether these three are equal, or what kind of relationship they are, requires a brief explanation.

    Thank you for the comment. Daidzein is a metabolite within the isoflavone class. The three are not separate things. I have rewritten this to clarify.

    1. There are few references on phytoestrogens in the last 5 years, and few studies on the effects of phytoestrogens on ruminants.

    There are numerous references included in the manuscript from 2015 onwards (21 journal articles and 2 industry reports)

    1. Part 8 is inconsistent with the overall logic of the article and does not closely follow the title.

    Thank you for the comment. I have removed part 8 to make the article consistent.

  1. The authors would like to thank you for the suggestions/comments. This has helped to improve the quality of the manuscript.